# School's out for summer–Differences in training characteristics between adolescent biathletes of different performance levels

**Andreas Kårström** [1,2☯]*, **Marko S. Laaksonen**[1], **Glenn Björklund**[1☯]

**1** Department of Health Sciences, Swedish Winter Sport Research Centre, Mid Sweden University, Östersund, Sweden, **2** Swedish Biathlon Federation, Östersund, Sweden

☯ These authors contributed equally to this work.
* andreas.karstrom@miun.se

**Data Availability Statement:** All relevant data are within the paper and its Supporting Information files.

## Abstract

The purpose of this study was to retrospectively describe the longitudinal changes of training variables in adolescent biathletes based on performance level. Thirty biathletes (15 men and 15 women) were included in the study and categorized as either national level biathletes (NLB, n = 21) or national team biathletes (NTB, n = 9). Retrospective training data was collected from training diary covering the biathletes' four years (Y1-Y4) as student-athletes at upper secondary school. Training data was divided into physical and shooting training variables. A linear mixed-effect model was used for comparing the difference of the performance group and year of upper secondary school on training characteristics. The NTB group achieved a greater annual training volume than the NLB group, especially during Y4 (594±71 $h \cdot y^{-1}$ vs 461±127 $h \cdot y^{-1}$, p < 0.001), through an increase in duration of each session and by completing more weekly training volume during the general phase (13.7±4.6 vs 10.0 ±4.9 $h \cdot w^{-1}$, p = 0.004). No difference was observed in relative training intensity distribution between the groups. The total number of shots fired was also greater for the NTB (9971 ±4716 vs 7355±2812 $shots \cdot y^{-1}$, p = 0.003). There was an equal frequency in illness and injury for both the NLB and NTB. Accordingly, the results of the present study describe longitudinal changes of biathlon training in adolescent biathletes that also may affect performance development.

## 1. Introduction

Biathlon is a winter sport that combines the physiologically challenging cross-country (XC) skiing with the fine motor skills of rifle marksmanship. To be a successful endurance athlete, such as a biathlete, the athlete must accumulate a large amount of physical training [1] along with sufficient recovery [2]. Previous studies have shown that XC skiing explains roughly 50% of the variation in biathlon performance in individual races [3] and ~ 60% in sprint biathlon races [3] while the remaining variation is explained by the shooting performance (shooting time and shooting results) [3, 4].

Biathlon training is performed using multiple types of training modes, for instance: running, roller skiing, cycling and on-snow skiing [5, 6].When monitoring physical training in

**Funding:** The author(s) received no specific funding for this work.

**Competing interests:** The authors have declared that no competing interests exist.

biathlon, the training dose is based on duration, mode of exercise and intensity. The intensity is usually scaled based on the percentage of maximum heart rate ($HR_{max}$) [7] or as the relationship between the heart rate (HR) and lactate values [5, 8, 9]. The duration at each intensity can be allocated either as the time in zone, based on the recorded time in different intensity zones; the session goal, based on the main intensity goal of a single session; or as a combination of the time spent in each zone and the main goal for the session [10, 11]. In general, as adolescent athletes age they are encouraged to train more systematically and with greater volume and specialization in order to prepare for more training and competition later in their career [12]. There are reasons to believe that a simple *copy-paste* strategy from a senior athlete's training structure is not optimal as it comes with a risk of injury or less than optimal development, due to the difference in training volume and biological maturation between adolescents and fully matured athletes [13, 14]. Endurance athletes in mid and late adolescence have been reported to exercise at least five times a week [15] or about 7 hours of endurance training [16]. Most late-adolescent athletes that aim to compete at an international level need to balance an athletic career and school simultaneously, through a double career in a school-sport system. Prior to upper secondary school (USS), most adolescent athletes choose to specialize, by choosing one sport to focus on while stopping practicing other sports, and to train and compete for > 8 months per year [12, 17]. Many USS provide resources for student-athletes to advance within a particular sport while simultaneously preparing for an academic career, for example through adequate facilities [18], coaches and adjusting the school schedule to training and competitions [19]. Many schools allow their students to extend their school attendance by one extra year to ease the double-career transition and maximize training. However only a fraction of athletes that try to maximize the outcomes of both a sporting career and an academic career actually succeed in doing so [17].

Given the limited knowledge of biathlon training in general and adolescent biathlon training in particular, the aim of this study was to retrospectively describe the longitudinal changes of training variables in adolescent biathletes based on performance level.

## 2. Methods

### 2.1 Subjects

Thirty biathletes (15 men and 15 women) were recruited, all of which were accepted as student athletes at a USS with a biathlon focus between the years of 2015 and 2019. Each subject was categorized into one of two different performance groups based on performance—either as national team biathletes (n = 9) or as national level biathletes (n = 21). At some point, either during or subsequent to their school years, those classified as NTB had been selected to the national team (development team and/or A-team) based on competition results, whereas the rest of the participants remained as NLB. Since data consisted of data on training that had already been performed, and no further intervention was carried out, no informed consent was obtained. However, all biathletes using the online training diary had agreed prior to registration that their training data may be used by the Swedish Biathlon Federation for the purpose of research. The study was approved by the Swedish Ethical Review Authority (Dnr: 2022-02826-01).

### 2.2 Experimental approach and periodization

This study used a retrospective cohort design for assessing the training of late-adolescent biathletes. In Sweden, there are six USS with a specialised biathlon focus. Admission to these educational institutions is competitive and the student-athlete has to demonstrate potential in

| Month | May | Jun Jul Aug Sep Oct | Nov Dec | Jan Feb Mar | Apr |
|---|---|---|---|---|---|
| Phase | Transition | General Preparation | Specific Preparation | Competition | Regeneration |
| Meso-cycle | 1 | 2-7 | 8-9 | 10-12 | 13 |

**Fig 1. The annual cycle described in training phases including time of year and duration of each phase.**

both their main sport and in school. USS is not compulsory but offers student-athletes an academic pathway if they want to apply for higher education after graduation.

Individual training data was collected with the approval of the Swedish Biathlon Federation. All data was retrieved from the training diary Maxpulse (Maxpulse, Johan Bergman, Östersund, Sweden) used by all student-athletes in the USS. Three criteria needed to be fulfilled for inclusion: 1) completed four years of Swedish USS with an additional focus on biathlon, 2) a consistently recorded training diary over four consecutive years and 3) continued to train to compete after USS for at least one year, to eliminate the risk that the training would be affected by a lack of motivation during the final year of school. A training year consisted of 13 mesocycles of four weeks per cycle, from the beginning of May until the end of April. Throughout the training year, the training shifted between designated specific training phases: transition (TRAN), general preparation (GP), specific preparation (SP), competition (COMP) and regeneration (REG). The timing of the periods and the duration of each period are illustrated in Fig 1 and are described in other studies on training characteristics in XC skiing [1, 20]. Each training year corresponded to a USS year: first year (Y1: age 16–17), second year (Y2; age: 17–18), third year (Y3: age 18–19) and fourth year (Y4: age 19–20).

## 2.3 Training monitoring

All training was planned using a modified session goal/time-in-zone approach, and was planned by a coach in an online training diary. The frequency of the planned training was counted and compared to the frequency of the performed training sessions. Training intensities were monitored according to the five-scale intensity scale used by the Swedish Biathlon Federation based on HRmax, previously described by Sylta et al. [11]. However, in the present study all training intensities were dichotomized to two intensity zones, (low intensity training (LIT) = zone 1–2, < 82% of HRmax and high intensity training (HIT) = zone 3–5, > 82% of HRmax), previously described by Tønnessen [1]. Illness and injury frequency were counted as training days affected per year.

The physical training characteristics were classified as session duration (h), training volume (h per week/phase/year), training frequency (sessions per week/phase/year) and training volume in either the specific or general mode. Training was categorized as either specific (e.g. classic and skating roller skiing, classic and skating on-snow skiing; both with and without rifle carriage) or general (e.g. running, cycling or "other").

Shooting training was divided into three different categories based on whether it was done during (i) LIT training, (ii) HIT training, including competitions or (iii) precision shooting training. The precision shooting training included zeroing before a session or competition, precision-score shooting, dry shooting without live ammunition and free shooting practice separate from a main endurance session. Precision shooting could be performed both with and without XC/roller skis. Shooting training was classified as the total number of shots fired per year and number of shots per training intensity.

## 2.4 Statistics

Data is presented as mean ± standard deviation (SD). The Shapiro-Wilk's test was used to test normality along with visual inspection of the residual plots. All statistical analyses were made using Jamovi (Jamovi, version 2.2.5, jamovi.org). A repeated measure design with a within-subject factor and a between-group factor was used using a linear mixed model. The model was fitted to assess the relationship between performance group and age group on the training characteristics. Performance group and age group were set as fixed effects and biathletes as a random effect, with random intercept across subjects. Each of the training characteristics was set as the dependent variable. A new statistical model was made for each of the variables. For training characteristics in different training phases, the performance group and phases were fitted as fixed effects. Due to the overall low number of participants, sex was not included as a fixed effect in any of the variables. The level of significance was set to $\alpha < .05$. When the main interaction showed statistical significance, post-hoc comparison was performed using the Bonferroni correction. Effect size was calculated as omega square ($\omega^2$) for the significant post-hoc test [21] and effect size values of 0.01, 0.06 and 0.14 were considered as small, medium and large effect, respectively [22].

# 3. Results

## 3.1 Physical training

The annual training volume, distribution of LIT and HIT, training specificity and number of performed sessions are all reported in Table 1. The NTB group accumulated ~18% more physical training compared to the NLB group during USS ($p < 0.01$, ES = 0.29), mainly explained by the extended session duration for the NTB ($p = 0.001$, ES = 0.26) and the fact that the NTB increased their training volume, especially during Y4 (594 ± 71 vs 461 ± 127, t(39.3) = -3.61, $p = 0.003$, ES = 0.34). Annual physical training volumes including volumes for LIT and HIT distribution are presented in Fig 2.

The relative specific training was greatest for both groups during COMP (~90%), with a 40% increase in specific training distribution from GP ($p < 0.001$, ES = 0.98) and a 10% increase from SP ($p < 0.001$, ES = 0.78). The NTB accumulated a greater volume of training for all specific training forms–on-snow skating skiing ($p < 0.001$, ES = 0.33) and classic skiing

**Table 1. Annual training characteristics for adolescent biathletes during four years of upper secondary school (mean ± SD).**

|  |  | Group | Y1 | Y2 | Y3 | Y4 |
|---|---|---|---|---|---|---|
| **Sessions performed (n)** | NTB | 311 ± 47 | 263 ± 23 | 293 ± 26 | 328 ± 33 | 359 ± 40 |
|  | NLB | 298 ± 64 | 253 ± 40 | 294 ± 45 | 327 ± 55 | 320 ± 84 |
| **LIT (%)** | NTB | 89 ± 2 | 87 ± 2 | 89 ± 1 | 90 ± 1 | 89 ± 1 |
|  | NLB | 89 ± 2 | 87 ± 3 | 87± 2 | 90 ± 1 | 90 ± 2 |
| **HIT (%)** | NTB | 10 ± 1 | 11 ± 2 | 10 ± 1 | 10 ± 1 | 11 ± 1 |
|  | NLB | 10 ± 2 | 11 ± 2 | 10 ± 2 | 10 ± 2 | 10 ± 2 |
| **Specific (%)** | NTB | 68 ± 27 * | 66 ± 31 | 70 ± 26 | 68 ± 27 | 69 ± 26 |
|  | NLB | 63 ± 29 | 63 ± 31 | 61 ± 30 | 62 ± 28 | 64 ± 28 |
| **General (%)** | NTB | 32 ± 27 * | 34 ± 31 | 30 ± 26 | 32 ± 27 | 31 ± 26 |
|  | NLB | 37 ± 29 | 36 ± 31 | 38 ± 30 | 38 ± 28 | 36 ± 28 |

LIT, Low-intensity training. HIT, High-intensity training. Y1, age group 16–17. Y2, age group 17–18. Y3, age group 18–19. Y4, age group 19–20. NTB, National team biathletes. NLB, National level biathletes

*Difference between national team biathletes and national level biathletes, p < 0.05.

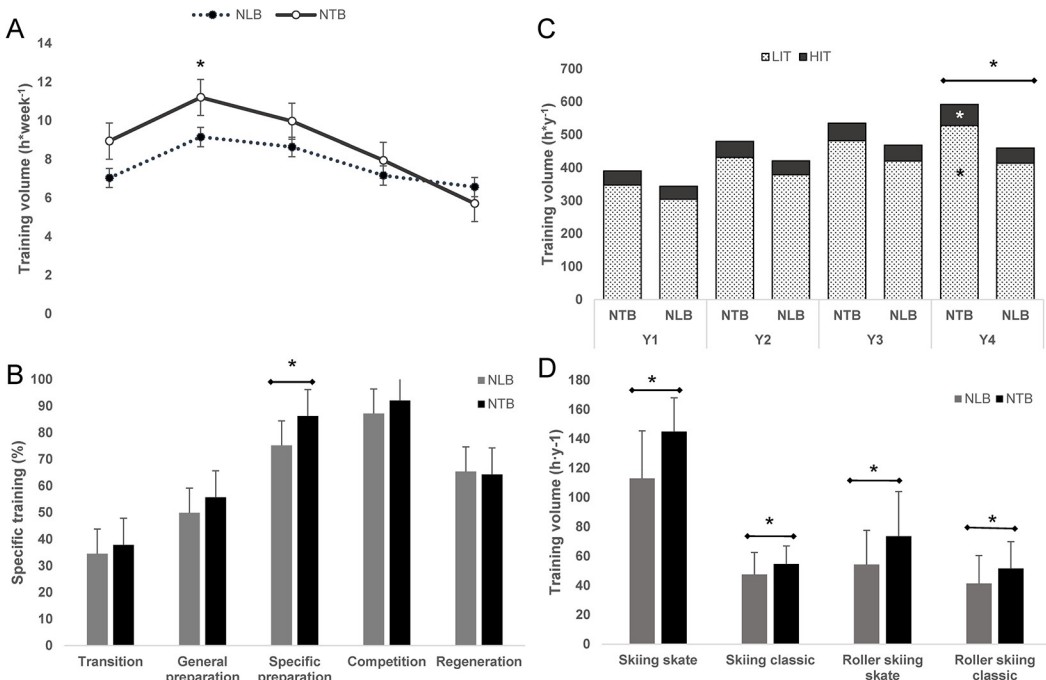

**Fig 2. Training volume and relative sport-specific training during different training phases and weeks for the national team biathletes and the national level biathletes.** (A) Training volume per week during different training phases. (B) Distribution of specific training in different training phases. (C) Annual training volume and intensity. (D) Annual training volume of specific training forms. * Difference between groups, p < 0.05. NTB, National team biathletes; NLB, national level biathletes.

(p = 0.013, ES = 0.16), roller skiing skating (p = 0.013, ES = 0.31) and classic (p = <0.001, ES = 0.16)–during USS, see Fig 2.

The NTB accumulated a greater weekly training volume during the GP phase compared to the NLB (p = 0.004, ES = 0.35), explained by more LIT performed by the NTB during this phase compared to the NLB (1.8 ± 0.40 h·week$^{-1}$, t(32) = -4.46, p = 0.004, ES = 0.39) while the HIT volume was similar (p = 0.255) between the groups. The difference in training volume by group during the GP phase is mainly explained by the training performed in Y4, when the NTB trained 27% more compared to the NLB during GP (p < 0.001, ES = 0.61). The NTB also had a higher proportion of specific training during the SP phase compared to the NLB (p = 0.008, ES = 0.34).

### 3.2 Shooting training

The total number of shots fired during training and competitions was greater for the NTB compared to the NLB (p = 0.003, ES = 0.22), mainly explained by the higher number of shots in both LIT and HIT (p < 0.001, ES = 0.45 and p < 0.001, ES = 0.34, respectively) between the two groups during Y4. The number of precision shots remained similar from Y1 to Y4 for both groups. The number of shots fired by NTB and NLB in different age groups is presented in Table 2.

### 3.3 Illness and injury data

An equal amount of training days was affected due to illness between the two groups (NLB: 20.5 ±16.8 days, NTB: 15.1 ±12.2 days, p > 0.106). Injuries did not occur more frequently in either of the two groups (NLB: 1.9 ± 5.0 days, NTB: 0.3 ± 0.8 days, p > 0.119).

**Table 2. Shooting characteristics for national team biathletes and national level biathletes across four age-groups at upper secondary school (mean ± SD).**

|  |  | Group | Y1 | Y2 | Y3 | Y4 |
|---|---|---|---|---|---|---|
| **Total shots (n)** | NTB | 9971 ± 4716 * | 8761 ± 4892 | 8961 ± 5645 | 10538 ± 4660 | 11624 ± 3682 * |
|  | NLB | 7355 ± 2812 | 5696 ± 2295 | 7634 ± 2487 | 8260 ± 3060 | 7828 ± 2817 |
| **Precision shots (n)** | NTB | 6166 ± 3582 | 5568 ± 3826 | 5512 ± 4105 | 6615 ± 3753 | 6968 ± 2953 |
|  | NLB | 4778 ± 2282 | 3438 ± 1777 | 4961 ± 2215 | 5629 ± 2512 | 5083 ± 2111 |
| **LIT shots (n)** | NTB | 2587 ± 1256 * | 2186 ± 1252 | 2361 ± 1539 | 2608 ± 1069 * | 3192 ± 1081* |
|  | NLB | 1742 ± 670 | 1505 ± 520 | 1811 ± 685 | 1774 ± 680 | 2005 ± 759 |
| **HIT shots (n)** | NTB | 1219 ± 524 * | 1007 ± 413 | 1089 ± 579 | 1315 ± 345 | 1464 ± 416 * |
|  | NLB | 835 ± 331 | 753 ± 290 | 862 ± 204 | 857 ± 345 | 867 ± 447 |

Y1, age 16–17. Y2, age 17–18. Y3, age 18–19. Y4, age 19–20. NTB, national team biathletes. NLB, national team biathletes. LIT, Low-intensity training. HIT, high-intensity training.

* Significant difference between groups, p < 0.05.

## 4. Discussion

This study is the first to retrospectively describe the longitudinal changes of training variables in adolescent biathletes based on performance level, in a structured training environment. The NTB had a greater volume of training than the NLB during USS. While the NTB managed to increase their training on a linear basis, the NLB did not further increase the volume of their training during Y4. NTB also showed a greater use of more sport specific training modes compared to NLB a well as a greater number of shots fired during both LIT and HIT.

The linear increase in training volume for NTB is mainly explained by the training they did during the GP, since the two groups did not differ in training volume over the other training phases. Given that the GP stretches over the summer months, the vast majority of the phase covers the summer vacation when no common structured training is performed. Consequently, more of the training needs to be self-motivated by the biathlete. Previous research has showed that athletes who successfully adhere to a training plan had a greater level of self-motivation compared to those who were less successful in adhering to the planned training [23]. Heavy training periods have been reported to increase the risk of mood disturbance in adolescent athletes, resulting in periods of staleness and loss of motivation [24]. Mood disturbance in athletes has previously been shown to increase linearly with training load [25]. Since the GP is generally considered to be the toughest training period in biathlon, the increased training load during this period could affect some biathletes negatively with mood disturbance and a loss of motivation to train. Other psychological factors such as the athlete-coach relationship have been suggested to play a major role for sport drop-out in adolescent athletes [26] and their ability to respond to training [27]. The desire to further develop in a main sport is also an important personal attribute for an athlete's motivation [28]. In a study by Karlsson et al., [20], adolescent national-team XC skiers aged 16 trained approximately 450 hours and then increased their training volume in a linear fashion over their final adolescent years. Since linear increase in the training volume seems to be a factor for further success in skiing sports, and both performance groups are part of the same school system, with the same opportunity to train equally for the majority of the time, the result of this study should be of interest to USS from a supportive perspective, as well as to coaches. The annual training volume reported by the NTB is similar to other studies examining the training characteristics of international-level biathletes [5, 8, 29].The distribution of ~ 90% LIT is similar to that of international senior-level athletes in XC-skiing and biathlon [1, 8]. Notably both the age and level of performance are considered greater in those papers [30]. The similarities in training characteristics between

studies despite the age and performance level can partly be explained by the variety of tools (e.g. training diary, method of reporting, technology etc.) used by athletes, coaches and associations to plan and monitor training.

The number of training sessions performed by late-adolescent athletes of different performance groups has previously been reported to not be a factor for further success [31]. In accordance with the present results, the greater training volume of the NTB with the same training frequency can be partly explained by the longer duration of each session, leading to more total accumulated training time. Training volume in biathlon is affected by the time spent on shooting practice, which gives less time to accumulate endurance training to the same extent when training is combined with shooting practice compared to unmitigated endurance training. This may indicate that the NTB assesses their physical training differently compared to the NLB, hence adding more physical training for compensating for the combined training.

The NTB had a greater volume of training in all specific training forms, showing the increased importance of sport-specific stimuli during both summer and winter conditions for selection as an NTB. The distribution of specific vs general training in the present study is similar to the previously reported results of a world-class XC skier [32]. It has previously been reported that highly skilled athletes accumulated more hours of structured training during late adolescence compared to less skilled athletes [33]. A high volume of year-round training with the primary aim to improve performance is the typical definition of sport specialization [34, 35]. Schooling in a structured training environment at a sport-focused USS should therefore be considered as an active sport-specialization. However, biathlon training is based on several different types of training forms [5, 6], diversifying the stimuli while still defining it as the main sport. In XC skiing, which has a very closely related training regime to biathlon, late adolescent XC skiers reported that 98% of their training is main-sport specific [36]. Previous studies have raised concerns about the increased risk of injury and psychological burnout from too-early sport specialization in adolescent athletes [13, 37], however the risk of injury from specialization in late-adolescent athletes is not as clear [38].

The greater volume of specific training simultaneously reflects the difference in number of shots fired in both LIT and HIT. Accordingly, the NTB spent a greater amount of their training on skis and thus fired more shots. In this study, the total number of shots fired by both groups is well under the reported number of shots reported in previous studies of international level biathletes [6, 8]. The vast majority (> 60%) of the shots in the present study were fired during precision exercises in various states of unloaded training. Precision shooting is often the back-bone of all shooting training with the goal to improve technical factors for accuracy and speed of preparation [6]. Technical factors such as stability of hold, shooting accuracy, triggering [39], minimal body sway and rifle movement [40, 41] are all associated with better shooting performance. However, shooting performance has been shown to decrease with elevated physical intensity. After high-intensity exercise, the stability of hold [42] along with postural balance and aiming accuracy [43] are all impaired. Thus, there is a need to train during conditions that are similar to those of a competition to improve performance. Greater skill acquisition in fine motor tasks induced by training has been shown to affect neural function and connectivity in the brain [44]. Since the NTB training consisted of significantly more shots fired during HIT compared to the NLB, prospective NTB could have improved their technical shooting skills during intense exercise and competitions. In previous studies, the number of missed targets has been shown to influence the final ranking at world cup competitions [45, 46]. Since selection to a national team is mainly based on competition results, the shooting accuracy during HIT and competitions is of great importance.

Illness and injury occurred equally frequently in both groups and did not explain the difference between the NTB and NLB. Differences in performance level have previously been

reported to not affect the risk of becoming ill or injured [38]. Adolescent biathletes seemed to be ill to the same degree as other adolescent winter athletes [20]. The prevalence of training cessation due to injury was not frequently reported in the present study (~1–2 days/year). One explanation could be that biathlon training can be fitted around many types of injuries through alternative training [47]. Types of injuries are not reported in this study, however previous studies of biathlon and XC skiing have shown that injuries are more frequent in the lower extremities compared to other body parts [47, 48].

There are some limitations that should be considered in the present study. The use of a new system for training planning and registration, together with self-reported data, should be considered when interpreting the data. The potential limitation of a retrospective study should be noted, as the data doesn't present any cause-effects relationship. The generalization that all biathletes will be of the same performance level if performing the same training may not be applicable. Furthermore, the dataset did not include other training forms such as strength or power that have been recorded in other studies of XC skiers [49, 50] due to the inconsistency on how to quantify these variables over this time period.

## 5. Practical application

This study has highlighted that it is important for adolescent biathletes to achieve a greater volume of training and to fire more shots during shooting practice, in order to increase their chance of being selected to a national team. This seems to be even more important during periods when the school system does not provide a structured training environment, such as school breaks. Adolescent biathletes that aim to be selected to a national team should also strive to increase their training volume on a linear basis over the full four-years of school attendance. This should mainly be done by increasing the duration of each session. The specific training should also be prioritized prior to the competition season. These findings should be applied by coaches and USS to create more suitable training plans and training conditions.

## Supporting information

**S1 File.**
(XLSX)

**S2 File.**
(XLSX)

## Acknowledgments

The authors would like to thank the Swedish Biathlon Federation for access to the data, as well as all participating biathletes and upper secondary schools. A special thanks goes to Gerold Sattlecker for his input to this manuscript. The authors have no conflict of interest to report.

## Author Contributions

**Conceptualization:** Andreas Kårström.

**Investigation:** Andreas Kårström.

**Supervision:** Glenn Björklund.

**Writing – original draft:** Andreas Kårström, Marko S. Laaksonen, Glenn Björklund.

**Writing – review & editing:** Andreas Kårström.

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
