## [Decision Letter · Decision Letter 0]

7 Jun 2023

PONE-D-23-04315School’s out for summer – Training characteristics of adolescent biathletes at upper secondary schoolPLOS ONE

Dear Dr. Kårström,

Thank you for submitting your manuscript to PLOS ONE. After careful consideration, we feel that it has merit but does not fully meet PLOS ONE’s publication criteria as it currently stands. Therefore, we invite you to submit a revised version of the manuscript that addresses the points raised during the review process.

Dear author,Study the comments of both reviewers and revise the manuscript in the light of suggestions of reviewers and submit revised journal accordingly.

We look forward to receiving your revised manuscript.

Kind regards,

Samiullah Khan, Ph. D

Academic Editor

PLOS ONE

Journal Requirements:

**Additional Editor Comments:**

Dear author,

Revise the manuscript in light of comments/suggestions of bot reviewers and submit the revised version accordingly.

Reviewers' comments:

Reviewer's Responses to Questions

**Comments to the Author**

1. Is the manuscript technically sound, and do the data support the conclusions?

Reviewer #1: Yes

Reviewer #2: Yes

2. Has the statistical analysis been performed appropriately and rigorously? 

Reviewer #1: Yes

Reviewer #2: Yes

3. Have the authors made all data underlying the findings in their manuscript fully available?

Reviewer #1: Yes

Reviewer #2: Yes

4. Is the manuscript presented in an intelligible fashion and written in standard English?

Reviewer #1: Yes

Reviewer #2: Yes

5. Review Comments to the Author

Reviewer #1: Good contribution to winter sports experimental study at upper school level.As in Sweden, there are six USS with a specialised biathlon focus. Admission to these educational institutions is competitive and the student-athlete has to demonstrate potential in both their main sport and in school. USS is not required, but offers student-athletes an academic background if they need to apply for higher education after graduation.This study is to examine late-adolescent biathletes in different performance groups based on their training characteristics, in a structured training environment. This study has highlighted that it is important for adolescent biathletes to achieve a greater volume of training and to fire more shots during shooting practice, in order to increase their chance of being selected to a national team. This study may help coaches and USS to create more suitable training plans and training conditions.

Reviewer #2: Comments to the Author

In this retrospective training study, the authors investigated training characteristics between late-adolescent Swedish biathletes of different performance groups The conclusion is that the best performing biathletes performed greater annual training volumes (with more progression) than their lower-performing peers, mainly caused by longer sessions in the general preparation period (when the schools are closed and thus having less structured training). Also, the best performing biathletes performed a greater volume (number of shots) of shooting training. Although the manuscript is well-written and of high relevance for (adolescent) biathlon training, I’m a bit concerned of the novelty and generalization of the findings. First, the findings can be considered rather “unsurprising” and second, such retrospective data does not provide any solid information (cause-effect) on how lower-performing biathletes actually should train to close the gap to higher performing biathletes. For this purpose, you need something prospective and ideally a training intervention to say something about actual training effects. Therefore, I have some questions and suggestions that needs to be addressed before the manuscript can be considered for publication.

Title

I’m not sure how good the title actually reflects the content of the study and it does not provide information on the different performance levels compared. Could it for example be something like “Differences in training characteristics between adolescent biathletes at different performance levels/of different performance groups”?

Abstract

L14: What is meant by seasonal effect? You are (IMO) not measuring the actual effect of something (providing cause-effect) but retrospectively describing differences in longitudinal training characteristics between biathletes of different performance levels. I think this must be revised throughout the manuscript to better suit your study design and related data.

L20: Again, I don’t like the word effect, could it be changes to influence of performance group and school?

L24: Is it possible to include some data here as well, not only p-value?

L25-26: Same here, is it possible to include actual data on the between-group differences, the p-value provides little information on its own.

L28-31: I find the conclusion a bit too unnuanced with too much generalization of the findings, although there are retrospectively observed differences between two groups of biathletes, it is not evident that the lower performing biathletes will close the gap to the higher performing biathletes by doing the same as they do. This can be included in the discussion with more focus on practical applications of the findings, but I suggest writing a conclusion that to a greater extent summarizes your findings rather then generalizing them too much into a practical context (particularly in the abstract).

Introduction

L42-44: The sentence “Low-intensity training (LIT) accounts for most of the physical training time in several endurance sports” seems a bit out of context considering the sentence before and after. I would have briefly summarized how biathletes train to meet the demands written above before finishing with the sentence that there are limiting training data in biathlon, and particular for adolescents.

L71-72: I would have changed the order here as the main emphasis is biathlon training. For example, “Given the limited knowledge of adolescent training and training of adolescent biathletes in particular, the aim of this study was”

L73-75: The purpose of the study here does not match the purpose written in the abstract (the seasonal effect). I suggest using (almost) the same purpose throughout the manuscript, and could it include something more “simple” like differences in training characteristics between adolescent biathletes at different performance levels/of different performance groups”?

Methods

L86-87: Didn’t the athletes provide any written consent to participate/share their data to the research project? This seems a bit strange..and unethical?

121: What was the rationale for using a two-zone intensity scale instead of a more nuanced 3-zone scale most common in retrospective training studies or just the original 5-zone scale to get an even more nuanced picture of the athletes intensity distribution?

L126: Is classical skiing and roller-skiing specific training for biathletes?

L129: is it possible to nuance/categorize the shooting training even more? There are many different parts of biathlon shooting training within the category precision shooting as it is now.

Results

General: You have included 15 men and 15 women in the study. Did you do any comparison between sexes for training characteristics and progression in training? Was the differences observed similar for both men and women? I think some sub-analyses on this would have strengthened your study, particularly considering that there are different development processes between sexes during such stages of their career.

Table1. I can not see the number of sessions planned as stated in L153?

The resolution of both Figure 1 and Figure 2 seems to be of low quality?

Discussion

Generally, the introduction is well written with several relevant for the study included. However, is it possible to have a more clear summary of the main findings, now there are many sentences and a bit hard to follow. Could also be useful with line spacing after the section summarizing the purpose and main findings.

L204-205: The purpose should be more consistent with the purpose in the abstract and introduction.

L205-206: The comparison of 90% LIT with previous literature in biathlon and xc skiing seem a bit out of context and should be moved to a later part of the discussion.

L293: Limitations, as stated above, I think the (potential) limitation of a retrospective design should be included. Although these data reveal interesting differences between biathletes of different performance levels, it does not provide any cause-effect relationship. Training and performance development is a complex process included by a variety of factors. Therefore, it is not evident (or it cannot be generalized) that lower performing biathletes will reach the same performance level as better performing ones by conducting the same training. Although it is likely that it will contribute positive to that if conducted with a sufficient level of progress and recovery etc. These aspects should be addressed in the discussion.

6. PLOS authors have the option to publish the peer review history of their article (what does this mean?). If published, this will include your full peer review and any attached files.

Reviewer #1: **Yes: **Dr Muhammad Zafar Iqbal Butt

Reviewer #2: No

---

## [Author Response · Author response to Decision Letter 0]

12 Jul 2023

Reviewer #2: Comments to the Author

In this retrospective training study, the authors investigated training characteristics between late-adolescent Swedish biathletes of different performance groups The conclusion is that the best performing biathletes performed greater annual training volumes (with more progression) than their lower-performing peers, mainly caused by longer sessions in the general preparation period (when the schools are closed and thus having less structured training). Also, the best performing biathletes performed a greater volume (number of shots) of shooting training. Although the manuscript is well-written and of high relevance for (adolescent) biathlon training, I’m a bit concerned of the novelty and generalization of the findings. First, the findings can be considered rather “unsurprising” and second, such retrospective data does not provide any solid information (cause-effect) on how lower-performing biathletes actually should train to close the gap to higher performing biathletes. For this purpose, you need something prospective and ideally a training intervention to say something about actual training effects. Therefore, I have some questions and suggestions that needs to be addressed before the manuscript can be considered for publication.

Response: Thank you for your time reviewing this manuscript and providing valuable feedback. We have reflected upon the insightful comments and have been able to incorporate changes on most of the reviewer’s suggestions. Changes in the manuscript are highlighted in bold and red text. Minor grammatical errors are corrected without notice. 

Title

I’m not sure how good the title actually reflects the content of the study and it does not provide information on the different performance levels compared. Could it for example be something like “Differences in training characteristics between adolescent biathletes at different performance levels/of different performance groups”?

Response: Thank you for this comment. The title is provocative with an intention since the majority of the difference between the two performance groups happens during the summer break from school. We have modified the title slightly and hope it is more straight forward and consistent with our findings 

Abstract

L14: What is meant by seasonal effect? You are (IMO) not measuring the actual effect of something (providing cause-effect) but retrospectively describing differences in longitudinal training characteristics between biathletes of different performance levels. I think this must be revised throughout the manuscript to better suit your study design and related data.

Response: Thanks for pointing this out. We have change it to retrospectively describe the longitudinal changes of training variables in adolescent biathletes based on performance level accordingly to your suggestion. (P2: L16-17)

L20: Again, I don’t like the word effect, could it be changes to influence of performance group and school?

Response: We agree with the reviewer and have changed the word effect to differences instead. (P2: L22)

L24: Is it possible to include some data here as well, not only p-value?

L25-26: Same here, is it possible to include actual data on the between-group differences, the p-value provides little information on its own.

Response: We understand the reviewers point and we have now included the data in both sections where there was previous only p-values (P2: L26-29) 

L28-31: I find the conclusion a bit too unnuanced with too much generalization of the findings, although there are retrospectively observed differences between two groups of biathletes, it is not evident that the lower performing biathletes will close the gap to the higher performing biathletes by doing the same as they do. This can be included in the discussion with more focus on practical applications of the findings, but I suggest writing a conclusion that to a greater extent summarizes your findings rather then generalizing them too much into a practical context (particularly in the abstract).

Response: Thank you for comment. We have tried to make the conclusion a clearer. (P2:L30-32)

Introduction

L42-44: The sentence “Low-intensity training (LIT) accounts for most of the physical training time in several endurance sports” seems a bit out of context considering the sentence before and after. I would have briefly summarized how biathletes train to meet the demands written above before finishing with the sentence that there are limiting training data in biathlon, and particular for adolescents.

Response: Thank you for pointing this out. That part of introduction is now scratched since it did not fit as well as intended. Instead there is a brief explanation about biathlon training (P2: L44-45). 

L71-72: I would have changed the order here as the main emphasis is biathlon training. For example, “Given the limited knowledge of adolescent training and training of adolescent biathletes in particular, the aim of this study was”

Response: Thank you for this suggestion. The aim has now been modified to include the word “biathlon” as suggested. (P4: L 71)

L73-75: The purpose of the study here does not match the purpose written in the abstract (the seasonal effect). I suggest using (almost) the same purpose throughout the manuscript, and could it include something more “simple” like differences in training characteristics between adolescent biathletes at different performance levels/of different performance groups”?

Response: Thank you for pointing this out. We have now revised this section, and it is also similar to the abstract. (P4: L72-73)

Methods

L86-87: Didn’t the athletes provide any written consent to participate/share their data to the research project? This seems a bit strange..and unethical?

Response: Each athlete who uses the training diary gives their permission that their training data is stored by the Swedish Biathlon Federation in accordance with GDPR and that their data can be of subject for evaluation and/or research used strictly confidential. We have not searched permission from each individual athlete to participate in this specific study, but have gotten the permission and access by the federation. Please see P:5 L84-86.

121: What was the rationale for using a two-zone intensity scale instead of a more nuanced 3-zone scale most common in retrospective training studies or just the original 5-zone scale to get an even more nuanced picture of the athletes intensity distribution?

Response: We understand the reviewers point of view, while the main reason to use LIT/HIT model is explained by the reason that during some years, it was not custom to train or monitor zone-3 training on all upper secondary school. Thereof, we choose the binary model presented by Tonnessen et al 2014. 

L126: Is classical skiing and roller-skiing specific training for biathletes?

Response: Thank you for a thoughtful question. We reasoned that both classic and skate skiing (on skis or roller skis) is a whole-body workout, and also sport specific physical training modes. It also helps when comparing our data with other studies that have combined classic and skate skiing as just skiing or roller-skiing.

L129: is it possible to nuance/categorize the shooting training even more? There are many different parts of biathlon shooting training within the category precision shooting as it is now.

Response: Due to the registration system in the training diary it’s not possible to further categorize the shooting. 

Results

General: You have included 15 men and 15 women in the study. Did you do any comparison between sexes for training characteristics and progression in training? Was the differences observed similar for both men and women? I think some sub-analyses on this would have strengthened your study, particularly considering that there are different development processes between sexes during such stages of their career.

Response: We have run the data with sex as a fixed effect, but there were no sex differences in any of the statistics. Further, due to the overall low number of participants we removed sex as a fixed effect. We understand the reviewer’s point of view and have added this information in the method section. (P7:L144-145)

Table1. I can not see the number of sessions planned as stated in L153?

Response: This was a typing error. Thank you for pointing this out!

The resolution of both Figure 1 and Figure 2 seems to be of low quality?

Response: Thank you for clarifying this. The files are in the correct format and dpi. 

Discussion

Generally, the introduction is well written with several relevant for the study included. However, is it possible to have a more clear summary of the main findings, now there are many sentences and a bit hard to follow. Could also be useful with line spacing after the section summarizing the purpose and main findings.

Response: Thank you for your feedback on the discussion. We have tried to make the main findings clearer and to use line spacing when appropriate. Please see P11, L208- 210. 

L204-205: The purpose should be more consistent with the purpose in the abstract and introduction.

Response: Thank you for addressing this issue. We have rewritten the purpose to match the purpose in the abstract. (P11, L204-205)

L205-206: The comparison of 90% LIT with previous literature in biathlon and xc skiing seem a bit out of context and should be moved to a later part of the discussion.

Response: This information has now been moved to a later section where it fits more appropriate to the surrounding texts. (P12: 234-236)

L293: Limitations, as stated above, I think the (potential) limitation of a retrospective design should be included. Although these data reveal interesting differences between biathletes of different performance levels, it does not provide any cause-effect relationship. Training and performance development is a complex process included by a variety of factors. Therefore, it is not evident (or it cannot be generalized) that lower performing biathletes will reach the same performance level as better performing ones by conducting the same training. Although it is likely that it will contribute positive to that if conducted with a sufficient level of progress and recovery etc. These aspects should be addressed in the discussion.

Response: Thank you for your feedback on this section. We have made the statements clearer for the reader and added text about the potential limitations related to the study design (P15: L297-300)

---

## [Decision Letter · Decision Letter 1]

8 Aug 2023

School’s out for summer – Differences in training characteristics between adolescent biathletes of different performance levels

PONE-D-23-04315R1

Dear Dr. Andreas ,

We’re pleased to inform you that your manuscript has been judged scientifically suitable for publication and will be formally accepted for publication once it meets all outstanding technical requirements.

Kind regards,

Samiullah Khan, Ph. D

Academic Editor

PLOS ONE

Additional Editor Comments (optional):

Reviewers' comments:

Reviewer's Responses to Questions

**Comments to the Author**

1. If the authors have adequately addressed your comments raised in a previous round of review and you feel that this manuscript is now acceptable for publication, you may indicate that here to bypass the “Comments to the Author” section, enter your conflict of interest statement in the “Confidential to Editor” section, and submit your "Accept" recommendation.

Reviewer #1: All comments have been addressed

Reviewer #2: All comments have been addressed

2. Is the manuscript technically sound, and do the data support the conclusions?

Reviewer #1: Yes

Reviewer #2: Yes

3. Has the statistical analysis been performed appropriately and rigorously? 

Reviewer #1: Yes

Reviewer #2: Yes

4. Have the authors made all data underlying the findings in their manuscript fully available?

Reviewer #1: Yes

Reviewer #2: Yes

5. Is the manuscript presented in an intelligible fashion and written in standard English?

Reviewer #1: Yes

Reviewer #2: Yes

6. Review Comments to the Author

Reviewer #1: The author has incorporated the changes suggested by the reviewer. This study retrospectively describes the longitudinal changes of training variables in adolescent biathletes based on performance level which may help upcoming athletes to improve their performance.

Reviewer #2: The authors have carefully considered all my comments adressed in the first revision of the manuscript and made changes accordingly. Therefore, I consider the manuscript as suitable for publication in its current form. Well done!

7. PLOS authors have the option to publish the peer review history of their article (what does this mean?). If published, this will include your full peer review and any attached files.

Reviewer #1: **Yes: **Prof. Dr. Muhammad Zafar Iqbal Butt

Reviewer #2: No

---

## [Editor Report · Acceptance letter]

16 Aug 2023

PONE-D-23-04315R1 

School’s out for summer – Differences in training characteristics between adolescent biathletes of different performance levels 

Dear Dr. Kårström:

I'm pleased to inform you that your manuscript has been deemed suitable for publication in PLOS ONE. Congratulations! Your manuscript is now with our production department. 

Kind regards, 

on behalf of

Dr. Samiullah Khan 

Academic Editor

PLOS ONE